# Cellulose Nanofiber Films and Their Vibration Energy Harvesting

**DOI:** 10.3390/s22166280

**Published:** 2022-08-21

**Authors:** Seok-Hyun Lee, Jaehwan Kim

**Affiliations:** 1Department of Electrical Engineering, Inha University, 100 Inha-ro Michuhol-ku, Incheon 22212, Korea; 2Department of Mechanical Engineering, Inha University, 100 Inha-ro, Michuhol-ku, Incheon 22212, Korea

**Keywords:** cellulose nanofiber, functional film, vibration energy harvesting

## Abstract

Cellulose, the most abundant sustainable material on Earth, has excellent mechanical and physical properties, high optical transparency, biocompatibility, and piezoelectricity. So, it has many possibilities for future materials, and many researchers are interested in its application. In this paper, cellulose nanofiber (CNF) and CNF/polyvinyl alcohol (PVA) films are made, and their vibration energy harvesting is studied. CNF was isolated by chemical and physical methods, and the CNF suspension was cast on a flat substrate to make a film. A cast CNF wet film stayed in a 5 Tesla superconductor magnet for 7 days, which resulted in CNF alignment perpendicular to the magnetic field. To further improve the mechanical properties of the CNF film, mechanical stretching was applied. The CNF suspension was mixed with PVA, giving the film toughness. The cast CNF/PVA wet film was mechanically stretched and dried, which improved the CNF alignment. The fabricated CNF and CNF/PVA films were characterized using scanning electron microscopy and X-ray diffraction to verify the alignment. By stretching, the aligned CNF/PVA film exhibits the largest mechanical properties along the aligned direction. The maximum Young’s modulus and tensile strength of the 50% stretched CNF/PVA film are 14.9 GPa and 170.6 MPa, respectively. Finally, a vibration energy harvesting experiment was performed by invoking the piezoelectric behavior of the pure CNF, and 50% stretched CNF/PVA films. The harvester structure was innovated by adopting a cymbal structure, which was beneficial to producing large in-plane strain on the films. The designed cymbal structure was analyzed using ANSYS, and its natural frequency was experimentally verified. The CNF/PVA film performs better vibration energy harvesting than the pure CNF film. The CNF/PVA film is applicable for biocompatible and flexible vibration energy harvesting.

## 1. Introduction

Recent advances in portable electronic devices and wireless sensor networks require limitless electrical power, and people have been searching for permanent portable power sources from the environment. Vibration energy harvesting (VEH) technology scavenges electrical energy from environmental vibration [1]. VEH is achieved using piezoelectric materials to convert vibration energy to electrical energy. Flexible piezoelectric materials are useful for VEH as they can be attached to arbitrary structures and even fabrics. Over the technical properties, behavioral and environmental characteristics of flexible piezoelectric materials, sustainability is becoming important for future materials [2]. Polyvinylidene fluoride, a well-known piezoelectric polymer, is a petroleum-based material that suffers from radiation and temperature variations. Sustainability is essential in future materials, and a flexible piezoelectric material that is environmentally friendly and renewable needs to be developed to hold sustainability above the piezoelectric characteristics of the material for VEH.

Cellulose is the plant fiber materials’ main chemical component and the pulp and paper’s most important and basic chemical composition [3]. Cellulose can be extracted from various natural resources; for example, bacteria, plants, wood, and algae. Cellulose obtained from nature has amorphous and crystalline parts. There are two kinds of nanocellulose: cellulose nanocrystal (CNC) and cellulose nanofiber (CNF). The CNF is fiber-like and generally has microns long with 4–20 nm cross-section. The CNC has a needle-shaped structure with 50–500 nm length and 3–10 nm width [4,5]. It is highly crystalline with a very high Young’s modulus ~150 GPa. Recently nanocellulose has been reported as a renewable piezoelectric material [6,7]. The uniformity and alignment of CNF are important to utilize its piezoelectric and mechanical properties [8,9,10]. An aligned CNF composite was made by electrospinning the CNF-polyvinyl alcohol (PVA) blend and depositing the CNF solution [11]. The composite exhibited flexibility, transparency, and enhanced piezoelectric and mechanical properties and a VEH was demonstrated. However, a long fabrication process is required for electrospinning and casting. 

In this paper, CNF piezoelectric films were fabricated by simply casting a CNF/PVA blend, followed by mechanical stretching and magnetic field alignment. Different ratios of CNF and PVA were considered, and a pure CNF film was also prepared for comparison. Once CNF suspension is cast, it is not suitable for stretching, and the dried film is so stiff that it is not suitable for VEH application. Thus, PVA was blended with CNF suspension. The prepared films were tested for VEH using a cymbal-type VEH device. The proposed CNF piezoelectric film is chemically safer than the regenerated cellulose piezoelectric paper since it does not use strong chemicals for dissolving cellulose [6,8]. The isolation of CNF, film fabrication and alignment by mechanical stretching and high magnetic field were illustrated, and their morphologies, mechanical, physical, and piezoelectric properties were investigated. The CNF was isolated by combining a chemical approach—2,2,6,6-tetramethylpiperidine-1-oxylradical (TEMPO)-oxidation and a physical approach—the aqueous counter collision (ACC) method [12]. A cymbal-type VEH device was demonstrated with the prepared films.

## 2. Materials and Methods

### 2.1. Materials 

Hardwood pulp, originally from Canada, was received from Chungnam National University in South Korea. Sodium bromide (NaBr, 99%), TEMPO (98%), sodium hypochlorite (NaClO, 15%), sodium chloride (NaCl, 98%), PVA (Mw 85,000–124,000, 99+% hydrolyzed), and hydrochloric acid (HCl, 37%) were purchased from Sigma Aldrich, Burlington, MA, USA. Sodium hydroxide anhydrous (NaOH, 98%) was bought from Daejung, South Korea. All reagents were of analytical grade and used as received without purification. 

### 2.2. CNF Isolation

CNF suspension was extracted using the TEMPO-oxidation and ACC treatments on the hardwood pulp [12]. The TEMPO-oxidation was done according to the previous work [13]. In brief, hardwood pulp was soaked in deionized (DI) water for one day and disintegrated using a food mixer for 10 min, followed by a high-speed digital homogenizer (T25 digital ultra Turrax, IKA, Staufen, Germany), 10,000 rpm and 10 min. Total 2400 g of suspension was made by mixing HW pulp 30 g, TEMPO 0.375 g, NaBr 3.75 g, NaClO 600 mL and DI water. When the reaction started, the pH was maintained at 12 by adding 0.5 M NaOH and 0.5 M HCl. A pH meter (Orion Star A211, Thermo Scientific, Beverly, MA, USA) was used to measure the pH. After essential time (30, 45, 60 min), the reaction was finished by adding 0.5 M HCl to adjust pH equal to 7 and adding ethanol 10 mL. Sodium and chloride ions were removed from the TEMPO-oxidized suspension by washing the suspension via a sieve (hole size = 90 μm) by flowing DI water 15 L five times. Since ACC equipment requires a homogeneous pulp suspension, further homogenization was performed using an ultrasonic homogenizer (SONPULS HD2200, Bandelin, Berlin, Germany) before the ACC treatment. The ACC treatment on the TEMPO-oxidized CNF suspension was made using the ACC machine (ACCNAC 100, CNNT, Suwon, Korea) for 30 passes. One pass means that 1 L of CNF suspension passes through the machine. As the pass increased, the CNF suspension turned transparent, indicating that CNF size decreased. As the number of passes increases, the crystallinity increases to improve the transparency and mechanical strength. Thus, the maximum number of passes, 30, was chosen. The ACC-treated CNF suspension was further fractionated using a centrifuge (Supra, 22K, Hanil Scientific, Kimpo, Korea) at 11,000 rpm and 1 h. Finally, 1.5 wt.% CNF suspension was obtained. Figure 1a represents the CNF isolation process. 

### 2.3. Film Preparation

Figure 1b shows the film preparation process. CNF suspension was deposited on a polycarbonate (PC) substrate. Since the PC surface is hydrophobic, the water-based CNF suspension cannot be attached to the PC. Thus, the PC surface was made hydrophilic by applying oxygen plasma treatment using oxygen plasma equipment (CUTE, Femto Science, Hwaseong, Korea). The CNF suspension was cast 2 mm thick on the plasma-treated PC substrate using a doctor blade. After drying in the air, the cast film was detached from the PC substrate by submerging it in an ethanol (90%) and DI water (10%) mixture.

Notice that the cast CNF film is not easy for mechanical stretching because the wet film is not pliable. Thus, the extracted CNF suspension was mixed with the PVA solution to allow stretchable behavior in the film. 10 wt.% PVA solution was prepared and blended with the 1.5 wt.% CNF suspension by vigorously stirring the mixture solution using a magnetic stirrer at 90 and 1 h. It was found that 50:50 of PVA and CNF suspension resulted in an optimum ratio for achieving the best mechanical properties. Thus, this mixing ratio for PVA and CNF suspension was kept. The CNF/PVA film was prepared via the same procedure previously mentioned. 

### 2.4. Film Alignment

CNF alignment in CNF and CNF/PVA films is essential for improving their mechanical and piezoelectric properties. There are various methods for improving the CNF alignment in the films: mechanical stretching [8,9,14], magnetic field application [15,16] and electric field application [16,17,18]. This research used two methods, namely mechanical stretching and magnetic field application.

House-made unidirectional stretching equipment was used for aligning CNF in the cast CNF/PVA films with different drawing ratios. Figure 2a shows the stretching equipment. The cast films were soaked in DI water again to maintain stretchable behavior, and both ends of the films were fixed on the equipment holders. The stretching speed was 0.05 mm/s, and the stretched films were dried under an IR heater for 1 h. The stretching ratio was changed from 10% to 50%. Note that the CNF/PVA film was stretchable, but the pure CNF film was hard to stretch. Thus, the mechanical stretching was applied to the CNF/PVA film only.

Since CNF is a diamagnetic material, it is aligned perpendicular to a magnetic field, but the field should be very high because of cellulose’s low magnetic permeability [15]. A high magnetic apparatus (HiMA, JASTEC, Fukupka, Japan) was used for the magnetic field applied to the CNF films. HiMA is a superconductor magnet that produces 5 Tesla magnetic field in its core, 180 mm in diameter. Figure 2b represents HiMA. The cast CNF films on the PC substrate were encapsulated in a petri dish, and the dish was situated horizontally in the core using plastic support for 1 week. Under the magnetic field, the CNF films were slowly dried and detached from the PC substrate. Since CNF is responsible for aligning in the magnetic field, and the pure CNF films have more CNF than the CNF/PVA film, the magnetic field application was performed on the pure CNF film only. 

### 2.5. Film Characterization

CNF and CNF/PVA film characterization were investigated by analyzing structural morphologies, piezoelectric behaviors, and mechanical properties before applying for VEH. Film morphology was investigated using a field emission scanning electron microscope (FESEM, S-4100, Hitachi, Tokyo, Japan). X-ray diffraction patterns were investigated using an X-ray diffractometer (X’pert Pro MRD, Malvern Panalytical, Malvern, UK). Two-dimensional wide-angle X-ray diffraction (2D WAXD) was taken by a high-resolution X-ray diffractometer (D8 Discover, Bruker with Vantec 500 detector, Madison, WI, USA). 40 kV and 40 mA of CuKα radiation source were used with 1.0 mm of beam diameter on transmission mode for 2500 s. 

Chemical interactions of films were investigated by taking Fourier Transform-Infrared (FT-IR) spectra using an FT-IR spectrometer (Cary 630, Agilent Technologies, Santa Clara, CA, USA). 

The mechanical properties of the films were tested using a tensile test machine [19] according to ISO 527-3 Standard. The tensile test machine is equipped with an environmental chamber, and the temperature and relative humidity were tuned to 25 °C and 40% RH. Tensile test specimens were cut into 10 mm × 70 mm, and the gauge length was 40 mm.

Piezoelectric properties of the films were characterized with the same tensile test equipment and a picoammeter (6485, Keithley, Solon, OH, USA). Both surfaces of the specimens were deposited with aluminum electrodes of 10 × 40 mm^2^ using a thermal evaporator (SHE-6D-350T, Samhan Vacuum Co., Paju, Korea) [15]. The prepared specimens were fully discharged to eliminate the accumulated space charges before the test. The longitudinal direction of the films matched the casting direction to improve the CNF alignment. Stress–strain curves and piezoelectric current outputs were acquired by LabVIEW software in the test system. A load cell (UU-K10, Dacell, Cheongwon-gun, Cheongju, Korea) and a linear scaler (GB-BA/SR128-015, Sony, Tokyo, Japan) measure the applied load and displacement. A quasi-static induced charge was calculated from the current measured from the specimens using the picoammeter during the tensile test. The piezoelectric charge constant, d_31_, can be determined by [19],
(1)$$d_{31}={\left(\frac{\partial D_{3}}{\partial T_{1}}\right)}_{E}=\frac{\mathrm{induced}\;\mathrm{charge}\;\mathrm{per}\;\mathrm{unit}\;\mathrm{electrode}\;\mathrm{area}}{\mathrm{applied}\;\mathrm{in}-\mathrm{plane}\;\mathrm{normal}\;\mathrm{stress}}\;\left[C/N\right]$$

During tensile and piezoelectric tests, the specimens were maintained at 25 °C temperature and 40% relative humidity (RH) in an environmental chamber.

### 2.6. Vibration Energy Harvester

The VEH device used a cymbal structure with prepared CNF and CNF/PVA films. Figure 3 shows the schematic of the cymbal-type VEH. The thermal evaporator coated both sides of the film with aluminum electrodes. The film was firmly attached to the cymbal structure, and the head mass was installed on the top of the structure [20]. Cymbal structure is beneficial for amplifying strain in the film. The head mass inertia generates a tensile force on the film, inducing electrical charges when the VEH is excited using a shaker (HEV-50, Eliezer, Seoul, Korea). The VEH test setup was provided on an optical isolation table (RT Series, Newport, Irvine, CA, USA) and vibration isolation columns (Stabilizer I-2000 Series, Newport, USA). A function generator (33220A, Agilent, Santa Clara, CA, USA) generates an excitation signal, fed to the shaker, producing vibration to the VEH. A force transducer (8230-001, Brüel & Kjær, Nærum, Denmark) was mounted between the cymbal structure and the shaker to measure the excitation force. The VEH’s induced voltage was measured using a pulse analyzer (3560-B-030, Brüel & Kjær, Denmark). The resonance frequency of the VEH can be found by sweeping the excitation signal frequency. The VEH performance was investigated at its resonance frequency.

## 3. Results

### 3.1. Isolated CNF

CNFs were isolated with different TEMPO-oxidation times (30, 45, 60 min) and a different number of the ACC pass (10, 20, 30 passes). It has been known that the CNF size reduces as the TEMPO-oxidation time and the ACC pass increase [12,13,21]. As a result, the CNF sample at 60 min TEMPO-oxidation and 30 pass ACC exhibited the smallest size. Figure 4a represents the atomic force microscope (AFM, Veeco 200, Plainview, NY, USA) image of the isolated CNF sample. No big-size CNF was observed. Figure 4b shows the size (length) distribution of the CNFs. The size was counted from the AFM image using Image J software. Its mean length is 614 ± 276 nm, and its mean width is 13.3 ± 4.0 nm. 

### 3.2. Prepared Films 

Figure 5a,b show photographs of the prepared pure CNF film (a) and CNF/PVA film (b). They are quite transparent: their transparencies are 87.2% and 89.3% at 500 nm wavelength (see Figure 5c). The prepared CNF suspension by 60 min TEMPO-oxidation and 30 pass ACC was used for the film preparation.

The chemical structure of the prepared films was examined by taking FTIR spectra (Figure 5d). The peak at 1027 cm^−1^ corresponds to C-O stretching in PVA and CNF. The 1423 cm^−1^ peak corresponds to C-C stretching, and a board band at 3320 cm^−1^ corresponds to OH bonded groups. The reinforcement of CNF with PVA shows an additional peak at 1321 cm^−1^, corresponding to C-C stretching bond. The band peak at 2924 cm^−1^ assigned to C-O stretching appeared when CNFs were added to the PVA matrix. The peak at 3320 cm^−1^ increased in the PVA-CNF nanocomposites due to the hydrogen bond. The FTIR spectra indicate that CNFs have interacted well with PVA chemically.

Figure 6 represents surface and cross-sectional SEM images of the pure CNF, CNF/PVA films and their images after the magnetic field application and mechanical stretching. The surface image of the pure CNF film (Figure 6a) is quite smooth, and a compact layered structure of CNFs is shown in Figure 6e. When the magnetic field was applied to the pure CNF film, the CNFs were aligned perpendicular to the magnetic field (M-field), shown in Figure 6b, maintaining a compact layered structure (Figure 6f). Figure 6c,g represent the surface and cross-sectional SEM images of the prepared CNF/PVA film. A smooth and compact layered structure is also shown. When 50% mechanical stretching was applied, CNFs were aligned with the stretching direction, as shown in Figure 6d. The more compact layered structure was also maintained after the stretching (Figure 6h).

Further structural characteristics of the films before and after alignments were investigated by taking XRD and 2D-WAXD patterns. Figure 7a shows the XRD patterns of the pure CNF and CNF/PVA films, the CNF film after magnetic field application, and the 50% stretched CNF/PVA film. The patterns were taken along the longitudinal (cast, aligned or stretched) direction. 16° and 22.5° peaks represent the (110) and (200) planes of cellulose I, and the 19.4° peak shows the orthorhombic structure of semi-crystalline PVA. The pure CNF film shows its (110) and (200) peaks clearly, and they were slightly decreased after the magnetic field application. However, the ratio between (110) and (200) peaks is the same, indicating that the crystallinity index is about the same. The XRD patterns of the CNF/PVA film show the (110) and (200) peaks and a new 19.4° peak for PVA. At 50% stretched, the (110) and (200) peaks increased, indicating improved CNF alignment. 

Figure 7b–e show 2D-WAXD patterns. Slight improved alignment of CNF is shown in the 2D-WAXD pattern of the magnetic field applied CNF film (Figure 7c) compared with the pure CNF film (Figure 7b). Figure 7d,e show the 2D-WAXD patterns of the CNF/PVA film before (Figure 7d) and after 50% stretching (Figure 7e). As can be seen, the stretching shows improved CNF alignment at the (200) peak in the 2D-WAXD pattern.

### 3.3. Mechanical and Piezoelectric Properties

Figure 8 shows stress-strain curves of the pure CNF, CNF/PVA, the magnetic field applied CNF, and 50% stretched CNF/PVA films. Table 1 shows the mechanical properties of the films extracted from the curves. Mechanical properties of the pure CNF film are comparable with the previously reported ones: Young’s modulus = 12.1 ± 0.9 GPa, tensile strength = 112.2 ± 15.9 MPa, strain-at-break = 1.2 ± 0.3% [22]. As the magnetic field was applied, the strain-at-break increased, which improved the tensile strength under the similar Young’s modulus. The pure CNF/PVA film exhibits good mechanical properties compared to the pure PVA (Young’s modulus = 0.123 GPa, tensile strength = 14.7 MPa, strain-at-break = 43%): Young’s modulus and tensile strength improved by 91 times and 6.7 times. These mechanical properties of the CNF/PVA film are much larger than the previous report [23]. Mechanical stretching was applied to the CNF/PVA film and the mechanical properties progressively improved as the stretching ratio increased [8]. Since the maximum stretching ratio of the CNF/PVA film was 50%, the mechanical properties at a 50% stretching ratio were included in Table 1. The maximum Young’s modulus and tensile strength of 14.9 GPa and 170.6 MPa were obtained from the 50% stretched CNF/PVA film. 

The piezoelectric properties of the prepared films were characterized according to the previous procedure [19], and the results are shown in the last column of Table 1. The piezoelectric charge constant, d_31_, of the pure CNF film was 4.8 ± 1.7 pC/N. When CNF was blended with PVA, its d_31_ was low to 2.0 ± 0.9 pC/N. These values are lower than the previously reported cellulose electroactive paper, made by regenerating and aligning cellulose, d_31_= 27 pC/N [20]. The self-standing film of CNF extracted from the birch wood exhibited its piezoelectric property of 4.7–6.4 pC/N [7]. The magnetic field alignment of CNF film did not show any meaningful improvement in d_31_. This might be due to the alignment of CNF along not only the longitudinal direction but also the CNF rotation away from the plane [24]. Similarly, magnetic field application on the ultrathin CNF film did not improve the piezoelectric property [23]. The mechanical stretching improved the piezoelectric charge constant of the CNF/PVA film. Mechanical stretching is a popular and easy way to align CNF films. 

### 3.4. Vibration Energy Harvesting

The prepared CNF and CNF/PVA films were applied for the VEH test. The copper head mass is 86 g, and the film size is 10 mm wide, 0.025 mm thick and 70 mm long. The cymbal structure material is stainless steel, and its width is 12 mm. The cymbal structure was designed before experimental testing using ANSYS software. ANSYS Workbench meshing program was used to mesh the structure with a 3D linear element, and the number of meshes is 15,000. Figure 9a represents the FEM model. The bottom of the cymbal structure was fixed, and mechanical properties of the prepared CNF/PVA film were used for the analysis. The resonance frequency of the model was found by performing harmonic analysis. As a result, the resonance frequency was 112 Hz (Figure 9b).

The VEH test was performed as shown in Figure 3. The resonance frequency was 110 Hz by frequency sweeping, which is very close to the simulation result. Figure 10 shows the voltage outputs of VEHs with CNF film and CNF/PVA film when 6 N force was applied at 110 Hz. 0.35 V_p-p_ and 0.8 V_p-p_ were obtained from the CNF and CNF/PVA film VEHs. These values are larger than the previous report [11]. The larger voltage output of the CNF/PVA film VEH might be associated with its large Young’s modulus. The CNF/PVA film is appropriate for biocompatible and flexible VEH. 

## 4. Conclusions

CNF and CNF/PVA films were prepared and characterized, and their vibration energy harvesting was studied. CNF was isolated by combining TEMPO-oxidation and the ACC method. Films were made by casting CNF suspension and CNF/PVA blends on a PC substrate. For aligning CNF, a cast CNF wet film stayed in a 5 Tesla superconductor magnet for 7 days, which aligned CNFs perpendicular to the magnetic field. To further improve the mechanical properties of the CNF film, mechanical stretching was applied to the CNF/PVA film. The cast CNF/PVA wet film was mechanically stretched and dried, which improved the CNF alignment. 

The fabricated CNF and CNF/PVA films were characterized using SEM, FTIR, XRD, and 2D-WAXD to verify the alignment. The films showed a compactly layered structure, and CNFs were aligned perpendicular to the magnetic field direction in the CNF film and to the stretching direction in the CNF/PVA film. XRD and 2D-WAXD patterns proved the CNF alignment made by mechanical stretching. The mechanical properties of the pure CNF film were comparable with the previously reported ones. The strain-at-break and tensile strength increased as the magnetic field was applied to the pure CNF film. The pure CNF/PVA film exhibited good mechanical properties compared to the pure PVA: Young’s modulus and tensile strength improved by 91 and 6.7 times, which are much larger than the previous report. The maximum Young’s modulus and tensile strength of 14.9 GPa and 170.6 MPa were obtained from the 50% stretched CNF/PVA film.

Finally, vibration energy harvesters were made with pure CNF and 50% stretched CNF/PVA films with the cymbal-type harvester structure. The designed cymbal structure was analyzed using ANSYS, and its natural frequency was experimentally verified. The CNF/PVA film showed better vibration energy harvesting performance than the pure CNF film. 0.8 V_p-p_ were obtained from the CNF/PVA film VEH. The CNF/PVA film can be used for biocompatible and flexible vibration energy harvesting.

## Figures and Tables

**Figure 1 sensors-22-06280-f001:**
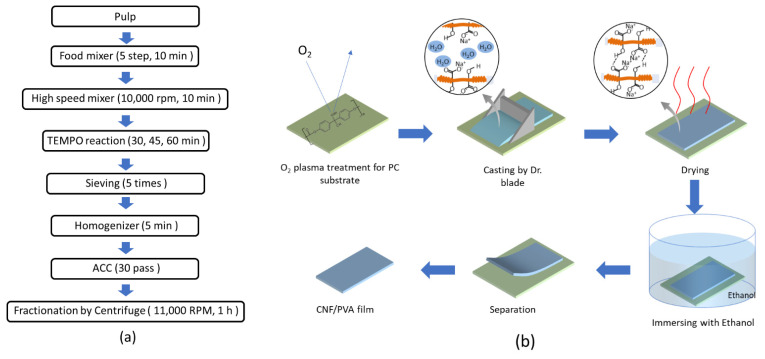
(**a**) The CNF isolation process made using TEMPO-oxidation and ACC methods; (**b**) CNF and CNF/PVA films fabrication process.

**Figure 2 sensors-22-06280-f002:**
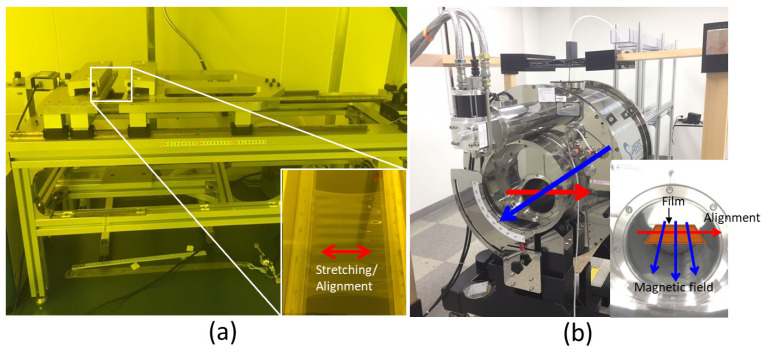
Photographs: (**a**) mechanical stretching system; (**b**) High magnetic apparatus. Red arrow lines indicate alignment direction, and blue ones indicate the magnetic field direction.

**Figure 3 sensors-22-06280-f003:**
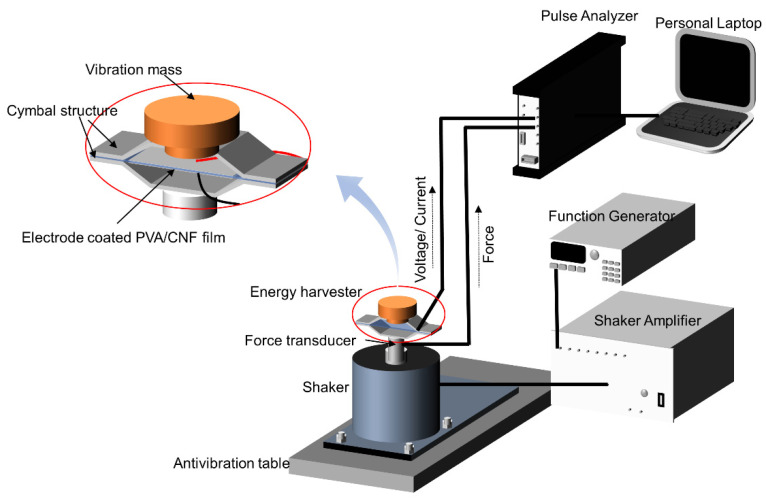
A schematic diagram of vibration energy harvesting and its test setup.

**Figure 4 sensors-22-06280-f004:**
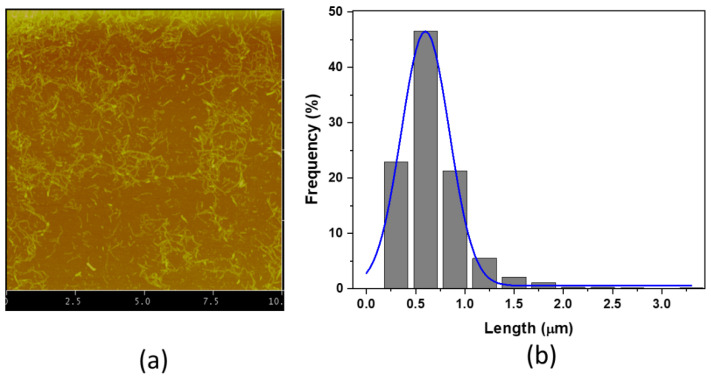
(**a**) An AFM image of the isolated CNF at 60 min TEMPO-oxidation and 30 ACC pass; (**b**) the size distribution in length.

**Figure 5 sensors-22-06280-f005:**
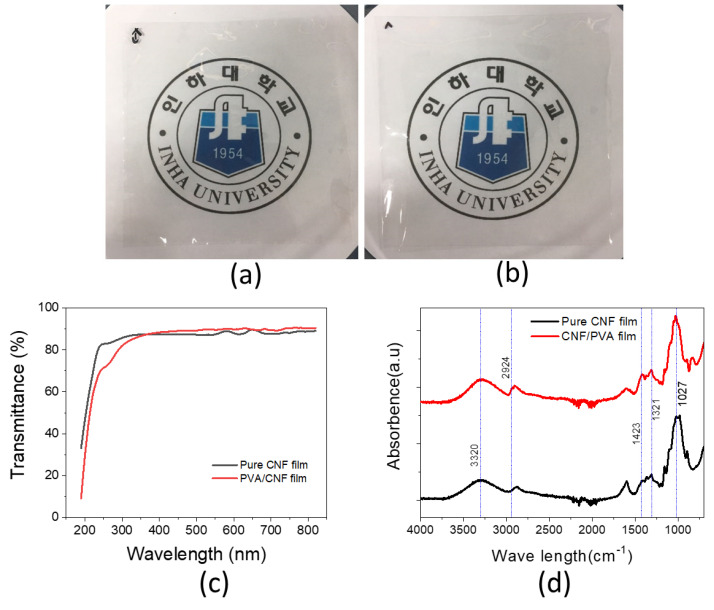
(**a**) Photographs of the pure CNF film; (**b**) a photograph of the CNF/PVA film; (**c**) UV-vis. Spectra; (**d**) FTIR spectra.

**Figure 6 sensors-22-06280-f006:**
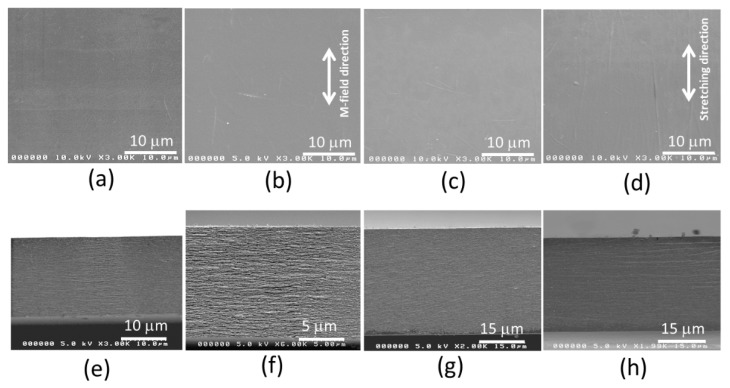
(**a**,**e**) Surface and cross-sectional SEM images of the pure CNF film (**a**,**d**); the magnetic-field applied CNF film (**b**,**f**); the pure CNF/PVA film (**c**,**g**); and the mechanically stretched CNF/PVA film (**d**,**h**).

**Figure 7 sensors-22-06280-f007:**
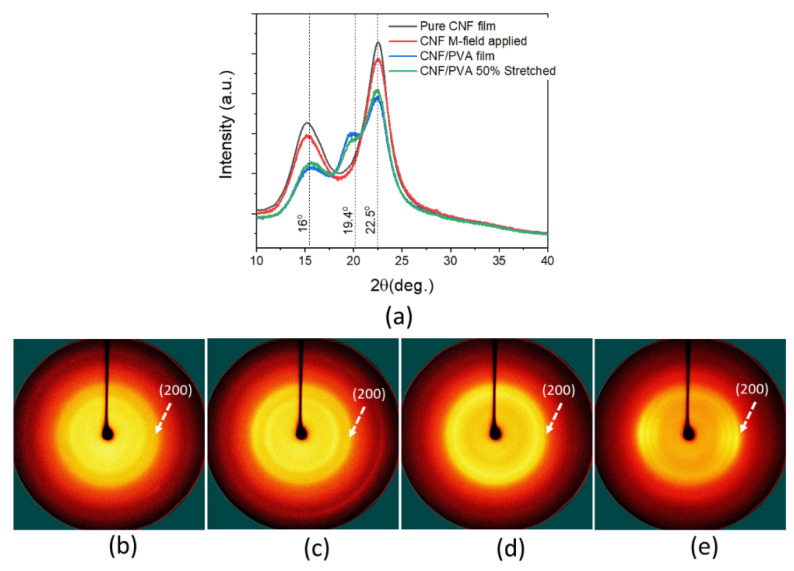
(**a**) XRD patterns of CNF and CNF/PVA films; 2-D WAXD patterns of (**b**) the pure CNF film; (**c**) the CNF film under M-field; (**d**) the pure CNF/PVA film, (**e**) the 50% stretched CNF/PVA film.

**Figure 8 sensors-22-06280-f008:**
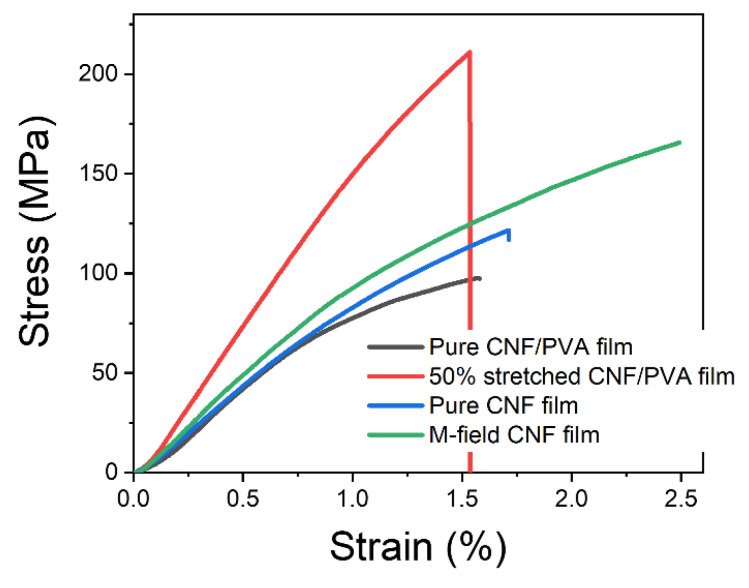
Stress–strain curves of prepared films.

**Figure 9 sensors-22-06280-f009:**
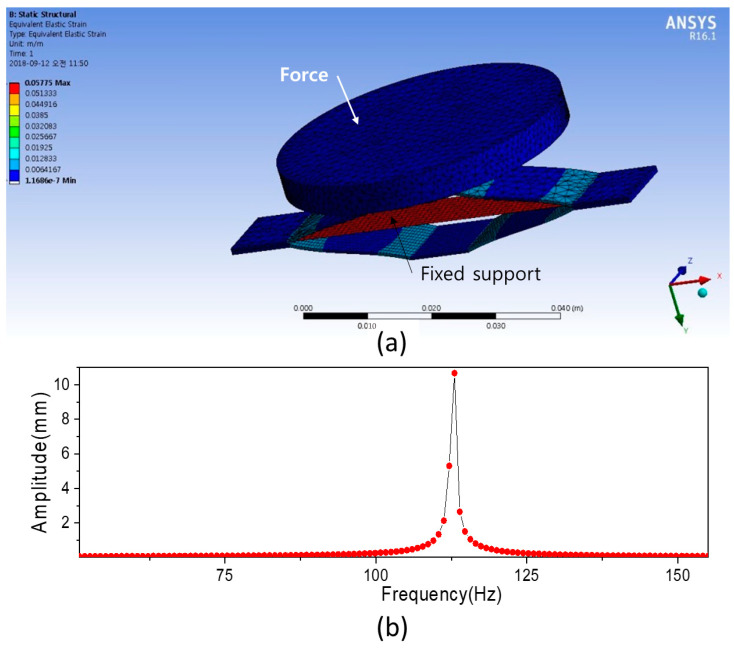
(**a**) FEM mesh model of VEH; (**b**) harmonic analysis result.

**Figure 10 sensors-22-06280-f010:**
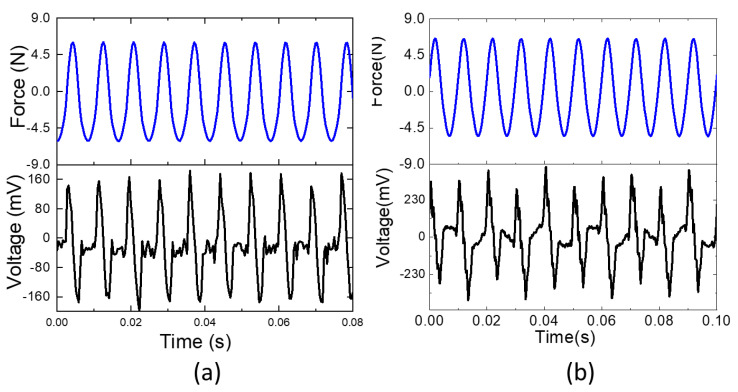
Input force and output voltage of (**a**) CNF film VEH; (**b**) CNF/PVA film VEH.

**Table 1 sensors-22-06280-t001:** Mechanical properties of the prepared CNF, CNF/PVA, the magnetic field applied CNF, and 50% stretched CNF/PVA films.

Film	Young’s Modulus (GPa)	Tensile Strength (MPa)	Strain-at-Break (%)	d_31_ (pC/N)
Pure CNF	10.1 ± 0.8	152.6 ± 43.9	2.3 ± 0.8	4.8 ± 1.7
M-field applied CNF	10.3 ± 0.7	164.6 ± 31.5	3.4 ± 1.9	3.7 ± 1.5
Pure CNF/PVA	11.2 ± 0.8	99.0 ± 3.3	1.5 ± 0.1	2.0 ± 0.9
50% stretched CNF/PVA	14.9 ± 1.5	170.6 ± 41.6	1.3 ± 0.2	4.4 ± 2.0

## Data Availability

Not applicable.

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
