# Peer review of "Cellulose Nanofiber Films and Their Vibration Energy Harvesting"

_sensors, 2022, doi:10.3390/s22166280_

Round 1

Reviewer 1 Report

In this work, CNF films were fabricated and applied to cymbal-type vibration energy harvesting device. Followings are comments for this paper.

1. The format of the presented figure seems to be different from the default format of the journal and needs to be corrected.

2. There are no titles for (c) and (d) in Figure 5.

3. Although the schematic is presented in Figure 3, it would be nice if a photo of the experimental device setting was added.

4. Is it possible to mass-produce CNF film for engineering applications?

Reviewer 2 Report

In this paper, cellulose nanofiber (CNF) and CNF/polyvinyl alcohol (PVA) piezoelectric films were fabricated for vibration energy harvesting. The isolation of CNF, film fabrication and alignment by mechanical stretching and high magnetic field were demonstrated, and their morphologies, mechanical, physical, and piezoelectric properties were investigated as well. This study is interesting and informative, but there are some issues that should be addressed before publication.

1. For vibration energy harvesting, the stability and repeatability of the device is very important. So how about the actual stability and repeatability of the functional film  after long-term operation? Please provide some experimental data.

2. Has the author studied the linear relationship between force and electricity? If so, please indicate

3.  What is the thickness of the composite film and why is this thickness used? 

4.  For vibration energy harvesting, the author should better study the matching of power with different load.

5.  Some descriptions conflict with English grammar and academic English writing norms. Please carefully proofread the manuscript and allow professionals to revise it.

Reviewer 3 Report

The manuscript by Lee et al. demonstrated the vibration energy harvesting devices based on cellulose/ PVA composites. By stretching and magnetic field alignment, the crystallinity and microstructure alignment are enhanced for the device application. The study was arranged in a quite good structure and I do not have much to comment on. I would suggest its publication after a minor revision. Please see my suggestions below.

·         Resolution of figures is low and high-quality figures are needed.

·         Scale bars are missing in Figure 7 b to e. In addition, the scale bars for Figure 6 e – h are different. Is there any explanation for this?

·         From figure 6f, I can see layer-by-layer type structure. What are the causes for this?

·         Is possible to estimate the crystallite size for each sample in Figure 7a?

·         Some parts are with typos or template problems. The Line 75 seems to have a large font size compared to other parts.

·         The part for vibration energy harvester is too short and limited, considering the title of the study. I would suggest adding more information in section 3.4, including device structure (which is now in the experiment part).

·         How is the power conversion efficiency for these devices? Would the low conductivity of cellulose/PVA influence the overall device efficiency?
